# The usefulness of D-dimer as a predictive marker for mortality in patients with COVID-19 hospitalized during the first wave in Italy

**Shermarke Hassan**[1,2], **Barbara Ferrari**[3], **Raffaella Rossio**[3], **Vincenzo la Mura**[1,3], **Andrea Artoni**[4], **Roberta Gualtierotti**[1,4], **Ida Martinelli**[4], **Alessandro Nobili**[5], **Alessandra Bandera**[1,6], **Andrea Gori**[1,6], **Francesco Blasi**[1,7], **Valter Monzani**[8], **Giorgio Costantino**[9,10], **Sergio Harari**[9,11], **Frits Richard Rosendaal**[2], **Flora Peyvandi**[1,3]*, **on behalf of the COVID-19 Network working group**[¶]

1 Department of Pathophysiology and Transplantation, University of Milan, Milan, Italy, 2 Department of Clinical Epidemiology, Leiden University Medical Center, Leiden, the Netherlands, 3 U.O.C. Medicina Generale Emostasi e Trombosi, Fondazione IRCCS Ca' Granda Ospedale Maggiore Policlinico, Milan, Italy, 4 Angelo Bianchi Bonomi Hemophilia and Thrombosis Centre, Fondazione IRCCS Ca' Granda Ospedale Maggiore Policlinico, Milan, Italy, 5 Department of Neuroscience, Istituto di Ricerche Farmacologiche Mario Negri IRCCS, Milan, Italy, 6 Infectious Disease Unit, Fondazione IRCCS Ca' Granda Ospedale Maggiore Policlinico, Milan, Italy, 7 Respiratory Unit and Cystic Fibrosis Adult Center, Fondazione IRCCS Ca' Granda Ospedale Maggiore Policlinico, Milan, Italy, 8 Acute Medical Unit, Department of Medicine, Fondazione IRCCS Ca' Granda Ospedale Maggiore Policlinico, Milan, Italy, 9 Department of Clinical Sciences and Community Health, University of Milan, Milan, Italy, 10 Emergency Department, Fondazione IRCCS Ca' Granda Ospedale Maggiore Policlinico, Milan, Italy, 11 UO di Pneumologia e Terapia Semi-Intensiva Respiratoria, Servizio di Fisiopatologia Respiratoria ed Emodinamica Polmonare, MultiMedica IRCCS, Milan, Italy

¶ The complete membership of the author group can be found in the Supplement
* flora.peyvandi@unimi.it

**Data Availability Statement:** All relevant data are within the paper and its Supporting Information files.

## Abstract

### Background

The coronavirus disease 2019 (COVID-19) presents an urgent threat to global health. Identification of predictors of poor outcomes will assist medical staff in treatment and allocating limited healthcare resources.

### Aims

The primary aim was to study the value of D-dimer as a predictive marker for in-hospital mortality.

### Methods

This was a cohort study. The study population consisted of hospitalized patients (age >18 years), who were diagnosed with COVID-19 based on real-time PCR at 9 hospitals during the first COVID-19 wave in Lombardy, Italy (Feb-May 2020). The primary endpoint was in-hospital mortality. Information was obtained from patient records. Statistical analyses were performed

**Funding:** The authors received no specific funding for this work.

**Competing interests:** B. Ferrari has received consulting fees and travel support from Sanofi Genzyme. R. Gualtierotti reports participation in advisory boards for Biomarin, Pfizer, Bayer and Takeda as well as participation at educational seminars sponsored by Pfizer, Sobi and Roche. I. Martinelli reports personal and non-financial support from Bayer, Roche, Rovi and Novo Nordisk outside of the submitted work. A. Gori has received grants for research support, honoraria, consultation fees, and travel support from Gilead, Janssen, MSD, Pfizer, Angelini, Menarini, ViiV. F. Peyvandi has received honoraria for participating as a speaker at educational meetings, symposia and advisory boards of Roche, Sobi, Sanofi, Grifols and Takeda. All other authors have no conflicts of interest to disclose. This does not alter our adherence to PLOS ONE policies on sharing data and materials.

using a Fine-Gray competing risk survival model. Model discrimination was assessed using Harrell's C-index and model calibration was assessed using a calibration plot.

## Results

Out of 1049 patients, 507 patients (46%) had evaluable data. Of these 507 patients, 96 died within 30 days. The cumulative incidence of in-hospital mortality within 30 days was 19% (95CI: 16%-23%), and the majority of deaths occurred within the first 10 days. A prediction model containing D-dimer as the only predictor had a C-index of 0.66 (95%CI: 0.61–0.71). Overall calibration of the model was very poor. The addition of D-dimer to a model containing age, sex and co-morbidities as predictors did not lead to any meaningful improvement in either the C-index or the calibration plot.

## Conclusion

The predictive value of D-dimer alone was moderate, and the addition of D-dimer to a simple model containing basic clinical characteristics did not lead to any improvement in model performance.

## Introduction

The coronavirus disease 2019 (COVID-19) is an urgent threat to global health that has severely strained the healthcare system of many countries [1–3]. Since the outbreak in early December 2019, the number of patients confirmed to have the disease has exceeded 521,563,472 and the number of people infected is probably much higher. More than 6,264,178 people have died from COVID-19 infection (up to May 16[th] 2022) [4].

Due to a large number of COVID-19 patients overwhelming the Italian healthcare system during the first COVID-19 wave in Lombardy, Italy (Feb-May 2020), it was important to look for biomarkers measured at admission that could predict mortality and other in-hospital adverse outcomes, in order to better triage patients.

D-dimer is a fibrin degradation product, which originates from the formation and lysis of cross-linked fibrin and reflects activation of coagulation and fibrinolysis. Among the clinical and biochemical parameters associated with poor prognosis, increased D-dimer levels seemed to be predictive for acute respiratory distress syndrome (ARDS), the need for admission to an intensive care unit (ICU) or death [5,6]. Furthermore, several studies have reported an increased incidence of thromboembolic events in hospitalized COVID-19 patients [7].

Taken together, these early studies indicate that D-dimer values at admission might be used to determine which patients would require hospitalization. Patients predicted to have a low enough mortality risk wouldn't need to be hospitalized, thereby decreasing the burden on the healthcare system.

Therefore, the primary aim of this paper was to study the predictive value of D-dimer levels at admission for in-hospital mortality. The secondary aim to assess if there was any causal relationship between D-dimer levels and in-hospital mortality.

## Methods

### Study design and population

This was an observational cohort study. The study population consisted of patients aged > 18 years who were hospitalized and who were positive for COVID-19 based on real-time PCR at 9 Italian hospitals, during the first COVID-19 wave in Lombardy, Italy (Feb-May 2020). Patients that were directly admitted to the ICU were excluded. Patients in this study were followed-up for 30 days from hospital admission.

This observational study was approved by the Medical Ethics Committee of the Fondazione IRCCS Ca' Granda Ospedale Maggiore Policlinico. The need to obtain informed consent was waived by the Medical Ethics Committee in cases where it was not possible to obtain informed consent, due to severe illness or death. In all other cases, written informed consent was obtained.

### Data collection and definition of variables

All information was obtained from electronic patient records, using a standardized case report form. The exposure of interest, D-dimer levels (expressed as ng/mL) was used as either a continuous variable or a categorical variable depending on the analysis of interest. When showing descriptive statistics and estimating the association between D-dimer levels and in-hospital mortality, D-dimer was converted to a categorical variable with 4 levels that correspond to the 1st, 2nd, 3rd and 4th quartile of D-dimer levels. This was done to make the results easier to interpret for the reader. D-dimer was analyzed as a continuous variable when assessing the predictive value of D-dimer levels at admission for in-hospital mortality. This is because categorizing a variable always leads to some loss in predictive power. The outcome used for all analyses was in-hospital mortality.

The following patient- and treatment characteristics were obtained: age (continuous variable), sex (dichotomous variable: male, female), the use of anticoagulant therapy during the study (dichotomous variable: yes, no) and the number of days between symptom onset and hospital admission (continuous variable). Lastly, the number of comorbidities (continuous variable) that each patient had at admission was calculated. Comorbidities to be assessed were selected based on their usage in the Charlson comorbidity index [8], a well-known risk score used to predict 10-year survival in patients with several comorbidities. To this list of comorbidities, we also added obesity (as defined by the clinician). The final list of comorbidities used was as follows: cardiovascular disease, chronic obstructive pulmonary disease, chronic kidney disease, diabetes mellitus, cancer, liver disease, dementia, connective tissue disease, acquired immunodeficiency syndrome (AIDS) and clinician-defined obesity.

### Statistical analysis: General approach

Descriptive analyses were reported as mean/standard deviation (SD), median/interquartile range (IQR), or as proportions. The cumulative incidence of in-hospital mortality and the relationship between D-dimer levels and in-hospital mortality were assessed using survival analysis methods. Discharge within 30 days with a good prognosis served as a competing outcome, in that it almost completely precludes the occurrence of the main outcome of interest (in-hospital mortality). Therefore, it was decided to model the relationship between D-dimer and in-hospital mortality using the Fine-Gray competing risk survival model, which accounts for the presence of competing events. In the multivariable analyses, we adjusted for age, sex and the number of comorbidities. For similar reasons, we did not use the Kaplan Meier function to estimate the cumulative incidence of mortality. Instead, we used the cumulative incidence

function, which correctly accounts for competing events. A complete case analysis was performed, meaning that patients with missing values for exposure, outcome or confounders were removed.

## Statistical analysis: Causal relationship between D-dimer levels and in-hospital mortality

The relationship between D-dimer levels (modeled as a categorical variable as described above) and in-hospital mortality was estimated using the aforementioned Fine-Gray competing risk survival model. We adjusted for age, sex and comorbidities, as well as anticoagulant therapy during hospitalization and the time between symptom onset and hospital admission.

## Statistical analysis: The predictive value of D-dimer for in-hospital mortality

D-dimer was modeled as a continuous variable for all analyses regarding the predictive value of D-dimer for in-hospital mortality. Three prediction models were tested: a model containing only D-dimer as a predictor, a model containing age, sex and comorbidities, and a model containing D-dimer, age, sex and comorbidities. For all three models, model discrimination and model calibration were assessed.

Model discrimination refers to how well a model can discriminate between patients with and without the outcome. Model discrimination was measured by calculating a modified version of Harrel's C-index [9], which is a measure of how well the model can discriminate between patients with and without the outcome. In the presence of competing risks, Harrel's C-index is biased [10]. We calculated a modified version of Harrel's C-index by setting the follow-up time of patients who experience a competing event to larger than our prediction horizon (which is 30 days), instead of censoring these patients, as was proposed by Wolbers et al. [10].

Model calibration refers to the degree to which the risk of mortality predicted by a model and the actual observed mortality rate in a group of patients are similar. For example, if the predicted mortality risk for a group of 100 patients is equal to 12% than the observed proportion of patients that died should also be around 12%. If the predicted mortality risk is much higher or lower than observed mortality rate, than the model is miscalibrated. To assess model calibration, we first fitted a model to the data, and then calculated the individual predicted mortality risk for each patient using this model. We then divided the patient population into ten groups (deciles) based on their predicted mortality risk. For each group, the predicted mortality risk for the whole group was compared with the observed mortality rate in that same group. This was done visually in a scatterplot that shows the predicted mortality risk on the X-axis and the observed mortality on the Y-axis for each decile. Furthermore, to examine calibration across the whole range, we also fitted a LOWESS (Locally Weighted Scatterplot Smoothing) line to the data.

## Sample size calculation

A formal sample size calculation for the development of a prediction model was not performed. However, 96 patients died during follow-up (see Results section) and the number of predictors used in the prediction models ranged from 1 (for the model containing only D-dimer as a predictor) to 4 (for the model containing D-dimer, age, sex and comorbidities as predictors). Accordingly, the number of events per predictor ranged from 96 to 24, well above

the minimum of 10 events per variable needed to accurately estimate the model coefficients [11]. Therefore, we deemed the sample size sufficient for these analyses.

## Results

### Baseline characteristics

Out of 1094 patients, 506 patients had missing D-dimer levels, 27 patients had incomplete follow-up data, 13 patients were excluded due to immediately being admitted to the ICU after admission and 41 patients were excluded due to missing data for one or more confounding factors. Finally, 507 patients (46%) had evaluable data. Of these, 96 patients died within 30 days after admission. Patients were enrolled between March 6th 2020 and September 20th 2020. Almost all (98%) patients were enrolled before May 31st 2020, which roughly corresponds to the first three months of the initial COVID-19 epidemic in Italy [12]. D-dimer values were associated with advanced age and the number of comorbidities at admission (Table 1).

From March 6th 2020 to September 20th 2020, the cumulative incidence of in-hospital mortality within 30 days was 19% (95CI: 16%-23%). Mortality was higher in the early phase of the epidemic and slightly decreased over time. (S1 Table) The cumulative incidence of discharge because of a good prognosis within 30 days was 72% (95CI: 68%-75%) (Fig 1) and 75% of deaths occurred in the first 10 days. After this period the death rate slowed down, as evidence by the flattening of the survival curve (Fig 1).

With increasing D-dimer levels, the absolute risk of mortality also increased strongly, from 4% (95CI:2%-9%) in patients with D-dimer levels in the lowest quartile to 28% (95CI: 20%-36%) in patients with D-dimer levels in the highest quartile (Table 2).

### Causal relationship between D-dimer levels and in-hospital mortality

Compared with patients in the lowest quartile of D-dimer blood concentration, the unadjusted hazard ratio for in-hospital mortality in patients in the 2nd, 3rd and 4th quartile was 4.5 (95CI: 1.7–11.8), 8.2 (95CI: 3.2–20.8) and 7.8 (95CI: 3.1–19.6) respectively. (Table 2) After adjusting for

**Table 1. Baseline characteristics of COVID-19 patients hospitalized in the region of Lombardy, Italy, during the first COVID-19 wave (Feb-May 2020).**

| Variables | D-dimer, < 538 ng/mL (N = 128) | D-dimer, 538–957 ng/mL (N = 125) | D-dimer, 957–1764 ng/mL (N = 127) | D-dimer, > 1764 ng/mL (N = 127) | Overall (N = 507) |
|---|---|---|---|---|---|
| mean age (SD) | 57.3 (16.4) | 62.3 (14.2) | 67.5 (14.1) | 69.3 (13.6) | 64.1 (15.3) |
| sex | | | | | |
| female | 46 (35.9%) | 31 (24.8%) | 47 (37.0%) | 52 (40.9%) | 176 (34.7%) |
| male | 82 (64.1%) | 94 (75.2%) | 80 (63.0%) | 75 (59.1%) | 331 (65.3%) |
| charlson comorbidity index | | | | | |
| no comorbidities | 68 (53.1%) | 70 (56.0%) | 64 (50.4%) | 52 (40.9%) | 254 (50.1%) |
| 1 comorbidity | 41 (32.0%) | 24 (19.2%) | 34 (26.8%) | 33 (26.0%) | 132 (26.0%) |
| 2 comorbidities | 15 (11.7%) | 21 (16.8%) | 14 (11.0%) | 28 (22.0%) | 78 (15.4%) |
| 3 or more comorbidities | 4 (3.1%) | 10 (8.0%) | 15 (11.8%) | 14 (11.0%) | 43 (8.5%) |
| anticoagulant therapy during hospitalization | | | | | |
| no | 28 (21.9%) | 25 (20.0%) | 18 (14.2%) | 25 (19.7%) | 96 (18.9%) |
| yes | 100 (78.1%) | 100 (80.0%) | 109 (85.8%) | 102 (80.3%) | 411 (81.1%) |
| mean number of days between first symptoms and admission (SD) | 8.9 (6.5) | 10.6 (12.3) | 11.7 (12.6) | 11.2 (10.5) | 10.6 (10.8) |

SD: Standard deviation.

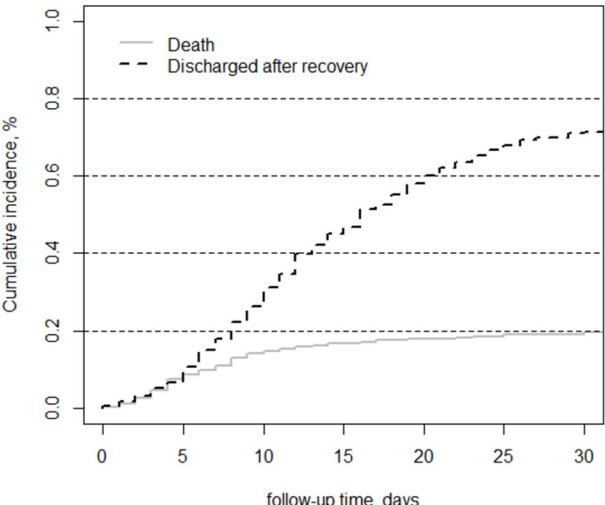

**Fig 1. Cumulative incidence function of 501 COVID-19 patients hospitalized in the region of Lombardy, Italy, during the first COVID-19 wave (Feb-May 2020).**

age, sex, comorbidities, anticoagulant therapy during hospitalization and the time between symptom onset and hospital admission, the hazard ratio for patients in the 2nd, 3rd and 4th quartile was 4.0 (95CI: 1.6–10.2), 5.4 (95CI: 2.1–13.8), and 4.5 (95CI: 1.8–11.5) respectively (Table 2).

## The predictive value of D-dimer for in-hospital mortality

The predictive model containing D-dimer as the only predictor had a C-index of 0.66 (95%CI: 0.61–0.71). Overall calibration of the model was very poor (Fig 2A). Next, the predictive model containing age, sex and comorbidities as predictors had a C-index of 0.83 (95%CI: 0.79–0.87). Overall calibration of the model was acceptable (Fig 2B). Lastly, the predictive model containing D-dimer, age, sex and comorbidities as predictors had a C-index of 0.83 (95%CI: 0.80–0.87). Overall calibration of this model was acceptable (Fig 2C).

## Discussion

Our results show that despite a strong correlation between D-dimer levels and mortality, the predictive value of D-dimer as a single biomarker was unclear. Model discrimination was

**Table 2. Association between D-dimer values and in-hospital mortality.**

|  | N | n | Observed incidence after 30 days | Univariate model | Multivariable model 1[a] | Multivariable model 2[b] |
|---|---|---|---|---|---|---|
| D-dimer |  |  |  |  |  |  |
| < 538 ng/mL | 128 | 5 | 0.04 (0.02–0.09) | ref | ref | ref |
| 538–957 ng/mL | 125 | 21 | 0.17 (0.11–0.24) | 4.5 (1.7–11.8) | 3.9 (1.5–9.9) | 4.0 (1.6–10.2) |
| 957–1764 ng/mL | 127 | 35 | 0.28 (0.21–0.36) | 8.2 (3.2–20.8) | 5.2 (2.0–13.1) | 5.4 (2.1–13.8) |
| > 1764 ng/mL | 127 | 35 | 0.28 (0.20–0.36) | 7.8 (3.1–19.6) | 4.3 (1.7–10.7) | 4.5 (1.8–11.5) |

[a] Model, corrected for age, sex and Charlson comorbidity index score.

[b] Model, corrected for age, sex, Charlson comorbidity index score, anticoagulant therapy during hospitalization, and the time between symptom onset and hospital admission.

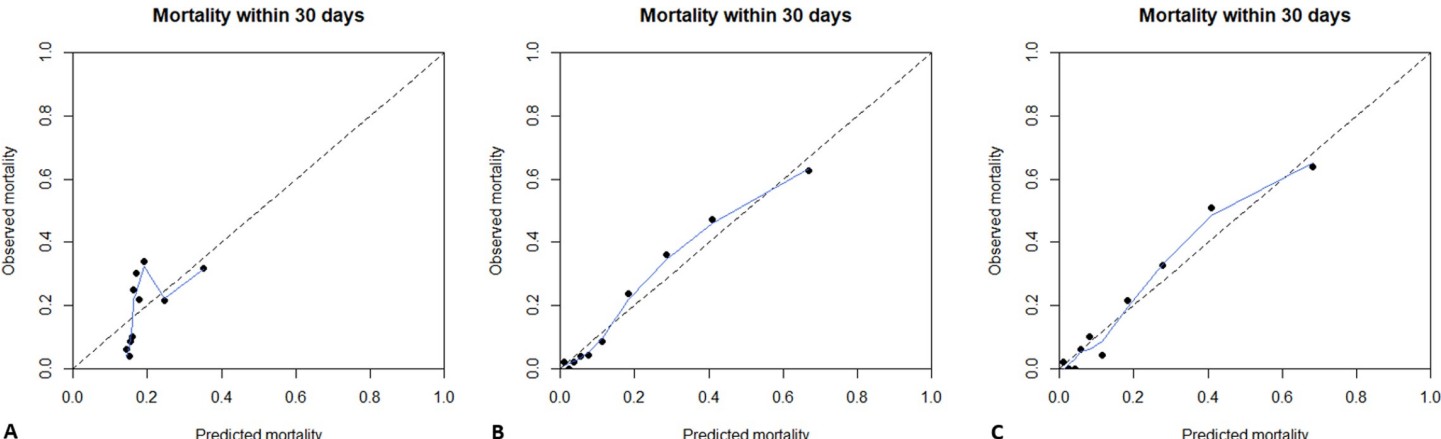

**Fig 2. Calibration plot of prediction models.** The figure shows the calibration plot of the model containing only d-dimer as a predictor (**A**), containing age, sex and comorbidities (**B**) and containing d-dimer, age, sex and comorbidities (**C**). The population was divided into ten groups (or deciles) based on their predicted mortality risk. (represented as black dots in the plot) The predicted probability of mortality according to the model is shown on the X-axis while the observed mortality is shown on the Y-axis. Groups with a higher predicted risk of mortality should have a higher observed risk. To examine calibration across the whole range, we also fitted a LOWESS (Locally Weighted Scatterplot Smoothing) line to the data. (shown here as a blue line) The dotted line represents perfect prediction (where the predicted risk is exactly the same as the observed risk).

moderate (C-index: 0.66) while model calibration was very poor. Furthermore, the addition of D-dimer to a simple model containing only basic clinical characteristics (age, sex and co-morbidities) did not lead to any meaningful improvement in either the C-index or the calibration plot.

D-dimer is a breakdown product, generated after a fibrin clot is degraded by fibrinolysis. It is a recognized valid lab biomarker that is widely used as part of the diagnostic workup of patients with a suspected venous thromboembolism or disseminated intravascular coagulation and is predictive of poor outcomes and thromboembolic events [13]. Changes in D-dimer levels are seen in most patients that are hospitalized with COVID-19 [14]. Changes in other hemostatic parameters, such as a slightly elongated PT, elongated aPTT, or mild thrombocytopenia are less common [15]. Furthermore, in addition to the increase in D-dimer (which is also an acute-phase protein that rises with general inflammation) an increase in inflammatory biomarkers such as CRP, particularly in COVID-19 patients with a more severe disease phenotype, is also seen [16].

Mechanisms underlying this COVID-19-induced coagulopathy may, in part, be explained by the same general mechanisms that also underlie other cases of bacteria-induced septic coagulopathy such as overproduction of pro-inflammatory cytokines by monocytes [17]. Furthermore, direct activation of coagulation by monocytes via tissue-factor and phosphatidylserine (which are expressed on the cell surface of monocytes) also play a role [16]. Furthermore, studies have reported endothelial dysfunction in patients with COVID-19 induced coagulopathy, which is probably mediated by the production of pro-inflammatory cytokines as well as activation of the complement cascade [18,19].

A strong correlation between D-dimer and mortality was also reported by other studies. A meta-analysis of six studies containing 1355 hospitalized patients found that D-dimer levels were higher in deceased patients (standardized mean difference: 3.59 mcg/L, 95%CI 2.79–4.40) [20]. This meta-analysis did not calculate a pooled C-index to assess the overall predictive performance of D-dimer.

A later meta-analysis reporting on 16 studies containing 4468 COVID-19 patients reported a pooled C-index of 0.86 (95CI: 83–89) for predicting all-cause mortality [21]. However, these

results were most likely strongly influenced by publication bias. In addition, it is somewhat unclear how the pooled C-index was calculated, as many studies did not report the C-index directly.

A retrospective study by Zhang et al. evaluated D-dimer levels and mortality in 343 patients [22]. D-dimer levels were measured within the first 24 hours, and hospitalized patients were followed until death or discharge. The study showed a very strong correlation between D-dimer levels over 2.0 mcg/mL and mortality. (HR: 51.5, 95%CI 12.9–206.7). However, the study did not adjust for any confounders. It is therefore unclear how different confounders could have affected the reported results. The predictive value of D-dimer was also very high (C-index: 0.89) There was no information about anticoagulant use during the study follow-up.

These strong results were not confirmed by a later study by Naymagon et al. [23] that followed 1062 COVID-19 patients during hospitalization. Each 1 μg/ml increase in D-dimer levels (measured within 3 days from admission) was associated with a hazard ratio of death of 1.05 (95%CI: 1.04–1.07). The association did not change after adjustment for age, smoking, Charlson comorbidity index and anticoagulant use at admission. However, discriminative performance of D-dimer levels was moderate (C-index: 0.694). At baseline, 9.1% of patients were on anticoagulants and no information was given about thromboprophylaxis during the study.

Overall, it seems that early studies reported that D-dimer was strongly predictive of mortality, although this effect was not as strong in the larger studies. Furthermore, all aforementioned studies only assessed the discriminative performance of D-dimer, but not model calibration.

Our study has several strengths. Firstly, our study had a large sample size with a sufficient number of events. Secondly, we applied a competing risk survival model to analyze the relationship between D-dimer and poor outcomes to avoid bias. Not taking competing risks into account could lead to misleading results, as shown in a recent simulation study on competing risks in COVID-19 research [24]. In addition, we evaluated both discrimination (as was done in earlier studies) and model calibration (which was not reported in any of the aforementioned studies). This is important because models may show good discrimination but could still be poorly calibrated [25].

Our study also has some limitations. The main limitation is that values for D-dimer levels were not available for 506 out of 1094 patients. D-dimer tests are most commonly ordered if a patient has some symptoms or medical history which are indicative of a thromboembolic event. Therefore, patients that were excluded from the study due to missing information on D-dimer were most likely patients with a low a priori likelihood of having a venous thromboembolic event. Also, D-dimer assays vary widely in their set-up. This lack of standardization makes comparison of different study results somewhat difficult [26,27].

Due to the rapid pace of change in the treatment of patients with COVID-19, the predictive value of D-dimer (and therefore, it's clinical usefulness) will most likely have diminished over time. For example, prophylactic anticoagulation for hospitalized COVID-19 patients became much more common over time. Furthermore, as the outbreak went on, patients with milder symptoms were also being hospitalized. Due to these treatment changes, we can speculate that patients hospitalized after the first COVID-19 wave will have had lower D-dimer levels at admission, when compared to patients admitted in the first COVID-19 wave (Feb-May 2020). Furthermore, D-dimer levels would have been less strongly associated with mortality in these patients, when compared to patients admitted in the first COVID-19 wave (Feb-May 2020).

As shown before, a part of COVID-19 related mortality is due to an underlying coagulopathy. (which might manifest as a venous thromboembolic event, as disseminated intravascular coagulation or as thrombotic microangiopathy) Consequently, some studies have suggested that D-dimer levels could be used to stratify patients with COVID-19, and to individualize treatment [22]. However, our analyses show that, despite a strong correlation between D-

dimer levels and mortality, the predictive value of D-dimer alone was not sufficient. However, that was to be expected as COVID-19 is not a coagulation disorder but a multi-systemic (although mainly respiratory) disease that influences health through multiple pathways, one of which is the coagulation system. However, D-dimer also showed little added value when added to simple risk prediction model containing only age, sex and comorbidities as predictors.

## Conclusions

The predictive value of D-dimer alone was moderate, and the addition of D-dimer to a simple model containing basic clinical characteristics did not lead to any improvement in model performance.

## Supporting information

**S1 Table. Cumulative incidence of death, per time-period.**
(DOCX)

**S1 List. List of participants in COVID-19 Network working group.**
(DOCX)

**S1 File. File containing the data and code used to generate the results.**
(ZIP)

## Acknowledgments

We thank the COVID-19 Network working group (contact: dr. Alessandro Nobili, alessandro.nobili@marionegri.it) for their help with patient recruitment and data collection. The full member list can be found in the Supplement.

## Author Contributions

**Conceptualization:** Barbara Ferrari, Raffaella Rossio, Vincenzo la Mura, Andrea Artoni, Roberta Gualtierotti, Ida Martinelli, Alessandro Nobili, Alessandra Bandera, Andrea Gori, Francesco Blasi, Valter Monzani, Giorgio Costantino, Sergio Harari, Frits Richard Rosendaal, Flora Peyvandi.

**Data curation:** Shermarke Hassan, Barbara Ferrari, Raffaella Rossio, Vincenzo la Mura, Andrea Artoni, Roberta Gualtierotti, Ida Martinelli, Alessandro Nobili, Alessandra Bandera, Andrea Gori, Francesco Blasi, Valter Monzani, Giorgio Costantino, Sergio Harari, Frits Richard Rosendaal, Flora Peyvandi.

**Formal analysis:** Shermarke Hassan.

**Methodology:** Shermarke Hassan.

**Writing – original draft:** Shermarke Hassan.

**Writing – review & editing:** Shermarke Hassan, Barbara Ferrari, Raffaella Rossio, Vincenzo la Mura, Andrea Artoni, Roberta Gualtierotti, Ida Martinelli, Alessandro Nobili, Alessandra Bandera, Andrea Gori, Francesco Blasi, Valter Monzani, Giorgio Costantino, Sergio Harari, Frits Richard Rosendaal, Flora Peyvandi.

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
