## [Decision Letter · Decision Letter 0]

8 Apr 2022

PONE-D-22-03309The usefulness of D-dimer as a predictive marker for mortality in patients with COVID-19 hospitalized during the first wave in Italy.PLOS ONE

Dear Dr. Hassan,

Thank you for submitting your manuscript to PLOS ONE. After careful consideration, we feel that it has merit but does not fully meet PLOS ONE’s publication criteria as it currently stands. Therefore, we invite you to submit a revised version of the manuscript that addresses the points raised during the review process.

We look forward to receiving your revised manuscript.

Kind regards,

Massimo Filippi

Academic Editor

PLOS ONE

Journal Requirements:

2. Thank you for including your consent statement: "Written informed consent was obtained from patients before data collection. In cases where it was not possible to obtain informed consent, due to severe illness or death. Data collection was still performed assuming the patient’s consent."

Please state in the ethics statement in the Methods and online submission form whether the ethics committee that approved your study waived the requirement for written informed consent from all patients.

(B. Ferrari has received consulting fees and travel support from Sanofi Genzyme. R. Gualtierotti reports participation in advisory boards for Biomarin, Pfizer, Bayer and Takeda as well as participation at educational seminars sponsored by Pfizer, Sobi and Roche. I. Martinelli reports personal and non-financial support from Bayer, Roche, Rovi and Novo Nordisk outside of the submitted work. A. Gori has received grants for research support, honoraria, consultation fees, and travel support from Gilead, Janssen, MSD, Pfizer, Angelini, Menarini, ViiV. F. Peyvandi has received honoraria for participating as a speaker at educational meetings, symposia and advisory boards of Roche, Sobi, Sanofi, Grifols and Takeda. All other authors have no conflicts of interest to disclose.)

We note that you received funding from a commercial source: (Sanofi Genzyme, Biomarin, Pfizer, Bayer, Takeda, Sobi, Roche, Bayer, Rovi, Novo Nordisk, Gilead, Janssen, MSD, Pfizer, Angelini, Menarini, ViiV, and Grifols)

4. One of the noted authors is a group or consortium (the COVID-19 Network working group). In addition to naming the author group, please list the individual authors and affiliations within this group in the acknowledgments section of your manuscript. Please also indicate clearly a lead author for this group along with a contact email address."

the COVID-19 Network working group

Reviewers' comments:

Reviewer's Responses to Questions

**Comments to the Author**

1. Is the manuscript technically sound, and do the data support the conclusions?

Reviewer #1: Yes

Reviewer #2: Partly

2. Has the statistical analysis been performed appropriately and rigorously? 

Reviewer #1: No

Reviewer #2: Yes

3. Have the authors made all data underlying the findings in their manuscript fully available?

Reviewer #1: No

Reviewer #2: Yes

4. Is the manuscript presented in an intelligible fashion and written in standard English?

Reviewer #1: Yes

Reviewer #2: No

5. Review Comments to the Author

Reviewer #1: The manuscript entitled “The usefulness of D-dimer as a predictive marker for mortality in patients with COVID-19 hospitalized during the first wave in Italy” by Shermarke Hassan et al., aims to demonstrate the value of D-dimer as a predictive marker for in hospital mortality in COVID-19 patients.

The manuscript addresses a simple and important question using a multicenter study involving a large sample size. The manuscript is well written.

There are some points that need to be addressed:

1. The statistical analyses need to be clearer.

Three prediction models were tested to show the causal relationship between D-dimer levels and in hospital mortality: a model containing only D-dimer as a predictor, a model containing

age, sex and comorbidities, and a model containing D-dimer, age, sex and comorbidities.

How was the calibration performed for each of the models using each of the quartiles?

The authors show the calibration plots in figure 2, but is not clear for these calibration plots which quartile of D-dimer are referring to?

Would be worth showing the calibration plot for each of their models developed using each of the quartiles.

Reviewer #2: The topic of the paper is timely and of clinical interest. Yet, I found the article to contain some incorrect statements and linguistic and grammatical errors. Thus, the article requires a complete revision. Specific comments:

-Page 2, section “Results”: add percentage of number of patients with evaluable data

-Page 3, section “Introduction”: numbers on people affected by COVID-19 and who died from the infection should be updated to April 2022

-Page 3, section “Introduction”, sentence “it was important to understand the role of early predictive markers”: please specify. Predictive of mortality? Disease worsening? Disability? Long-term consequences?

-Page 3, section “Introduction”: please rephrase the sentence “thereby decreasing the burden on the healthcare system” and avoid brackets

-Page 3, section “Introduction”, sentence “The secondary aim of this paper”: please remove the words “of this paper”, as they are a repetition

-Page 3, section “Methods-Study design and population”: please specify the starting time point of follow-up period: follow-up was of 30 days starting from hospital admission? Viral infection confirmed by PCR result?

-Page 3, section “Methods-Study design and population”: please rephrase the sentence “In cases where it was not possible to obtain informed consent […] assuming the patient’s consent.” Grammar is not correct; please avoid the full stop between the words “death” and “Data collection”

-Page 4, section “Methods-Data collection and definition of variables”: please remove the sentence “The primary endpoint […] mortality”, as endpoints have already been previously described

-Page 4, section “Methods-Data collection and definition of variables”: please define the Charlson comorbidity index, add reference

-Page 4, section “Methods-Data collection and definition of variables”, sentence “The total list of comorbidities was as follows;”: please change semicolons with colons

-Page 4, section “Methods-Data collection and definition of variables”, words “HIV aids”: please specify HIV infection and AIDS

-Page 4, section “Statistical analysis, general approach”: please define abbreviations before using them (e.g. SD, IQR…)

- Page 4, section “Statistical analysis, general approach”, sentence “in that it (practically) precludes”: please rephrase avoiding the use of brackets

- Page 4, section “Statistical analysis, general approach”, sentence “patients with missing values for the exposure”: please remove “the”

-Page 5, section “Statistical analysis, the predictive value of D-dimer for in-hospital mortality”: please replace semicolons with colons

-Page 5, section “Statistical analysis, the predictive value of D-dimer for in-hospital mortality”, paragraph “Model calibration was measured […] line to the data”: the whole paragraph is not clear and it is hard to follow. Could you please explain it better?

-Page 5, section “Results – Baseline characteristics”, sentence “either anticoagulant […] and hospital admission”: please rephrase this sentence, as English grammar is not correct

- Page 5, section “Results – Baseline characteristics”, sentence “between March 6th 2020 to September 6th 2020”: please replace TO with AND

-Page 5, section “Results – Baseline characteristics”, sentence “Almost all […] COVID-19 wave”: please rephrase avoiding the use of brackets

- Page 6, section “Results – Baseline characteristics”, sentence “Who died doing so”: please rephrase

- Page 6, section “Results – Baseline characteristics”: could you please specify how quartiles are defined and which are D-Dimer levels?

-Page 7, section “Discussion”, sentence “The mechanisms underlying this COVID-19 induced coagulopathy”: please remove “the” and add a “-“ between COVID-19 and induced (COVID-19-induced coagulopathy”;

- Page 7, section “Discussion”, sentence “The mechanisms underlying this COVID-19 induced coagulopathy may […] by monocytes”: add a reference explaining mechanisms of other bacteria-induced septic coagulopathies

-Page 7, section “Discussion”, sentence “measured within 3 days of admission”: please substitute “of” with “from”

-Page 8, section “Discussion”, sentence “9.1% of patients were on anticoagulant use […] during the study”: please rephrase avoiding the repetition of “anticoagulant use”

-Page 8, section “Discussion”: please define abbreviations: VTE, DIC, TMA…

-Page 8, section “Discussion”, sentence “For example in Lombardy […] were already being prescribed anticoagulant treatment”: please rephrase

-As suggested also by Authors, an important limitation of this study is the absence of D-Dimer levels in more than 500 patients included in the study, which represent approximately half of the analyzed cohort. Moreover, D-dimer is a non-specific marker of inflammation, therefore it is likely to be high in COVID-19 infected patients, regardless of the presence of coagulopathy and/or infection-related thrombotic event.

6. PLOS authors have the option to publish the peer review history of their article (what does this mean?). If published, this will include your full peer review and any attached files.

Reviewer #1: No

Reviewer #2: No

---

## [Author Response · Author response to Decision Letter 0]

7 Jun 2022

Editor

Request: 1. When submitting your revision, we need you to address these additional requirements. Please ensure that your manuscript meets PLOS ONE's style requirements, including those for file naming. The PLOS ONE style templates can be found at https://journals.plos.org/plosone/s/file?id=wjVg/PLOSOne_formatting_sample_main_body.pdf and https://journals.plos.org/plosone/s/file?id=ba62/PLOSOne_formatting_sample_title_authors_affiliations.pdf

Reply: I have applied the PLOS ONE style requirements

Request: 2. Thank you for including your consent statement: "Written informed consent was obtained from patients before data collection. In cases where it was not possible to obtain informed consent, due to severe illness or death. Data collection was still performed assuming the patient’s consent." Please state in the ethics statement in the Methods and online submission form whether the ethics committee that approved your study waived the requirement for written informed consent from all patients.

Reply: I have amended the consent statement as follows:

“This observational study was approved by the Medical Ethics Committee of the Fondazione IRCCS Ca' Granda Ospedale Maggiore Policlinico. The need to obtain informed consent was waived by the Medical Ethics Committee in cases where it was not possible to obtain informed consent, due to severe illness or death. In all other cases, written informed consent was obtained.”

I have also changed this in the online submission form.

Request: 3. Thank you for stating the following in the Competing Interests section: (B. Ferrari has received consulting fees and travel support from Sanofi Genzyme. R. Gualtierotti reports participation in advisory boards for Biomarin, Pfizer, Bayer and Takeda as well as participation at educational seminars sponsored by Pfizer, Sobi and Roche. I. Martinelli reports personal and non-financial support from Bayer, Roche, Rovi and Novo Nordisk outside of the submitted work. A. Gori has received grants for research support, honoraria, consultation fees, and travel support from Gilead, Janssen, MSD, Pfizer, Angelini, Menarini, ViiV. F. Peyvandi has received honoraria for participating as a speaker at educational meetings, symposia and advisory boards of Roche, Sobi, Sanofi, Grifols and Takeda. All other authors have no conflicts of interest to disclose.)

We note that you received funding from a commercial source: (Sanofi Genzyme, Biomarin, Pfizer, Bayer, Takeda, Sobi, Roche, Bayer, Rovi, Novo Nordisk, Gilead, Janssen, MSD, Pfizer, Angelini, Menarini, ViiV, and Grifols)

Within this Competing Interests Statement, please confirm that this does not alter your adherence to all PLOS ONE policies on sharing data and materials by including the following statement: "This does not alter our adherence to PLOS ONE policies on sharing data and materials.” (as detailed online in our guide for authors http://journals.plos.org/plosone/s/competing-interests). If there are restrictions on sharing of data and/or materials, please state these. Please note that we cannot proceed with consideration of your article until this information has been declared. Please include your amended Competing Interests Statement within your cover letter. We will change the online submission form on your behalf.

Reply: I am not sure if I understood this request correctly. What is meant by “Please provide an amended Competing Interests Statement that explicitly states this commercial funder”? Isn’t this exactly what I have already done? (i.e. mentioning these commercial parties in the competing interest statement) Also, what is meant by “We note that you received funding”? Do you mean funding for this project? None of the commercial parties mentioned in the competing interest statement have funded the current work. Could you please explain to me what further action needs to be taken? For now, I have added "This does not alter our adherence to PLOS ONE policies on sharing data and materials.” to the Competing Interest statement:

“B. Ferrari has received consulting fees and travel support from Sanofi Genzyme. R. Gualtierotti reports participation in advisory boards for Biomarin, Pfizer, Bayer and Takeda as well as participation at educational seminars sponsored by Pfizer, Sobi and Roche. I. Martinelli reports personal and non-financial support from Bayer, Roche, Rovi and Novo Nordisk outside of the submitted work. A. Gori has received grants for research support, honoraria, consultation fees, and travel support from Gilead, Janssen, MSD, Pfizer, Angelini, Menarini, ViiV. F. Peyvandi has received honoraria for participating as a speaker at educational meetings, symposia and advisory boards of Roche, Sobi, Sanofi, Grifols and Takeda. All other authors have no conflicts of interest to disclose. This does not alter our adherence to PLOS ONE policies on sharing data and materials.”

Request: 4. One of the noted authors is a group or consortium (the COVID-19 Network working group). In addition to naming the author group, please list the individual authors and affiliations within this group in the acknowledgments section of your manuscript. Please also indicate clearly a lead author for this group along with a contact email address."

Reply: I have added the information of the lead author of this group to the acknowledgements section. I have also added the full list of authors and affiliations, but because it is very long, I have added it as a supplement instead of putting it in the acknowledgements. 

Reviewer 1

The manuscript entitled “The usefulness of D-dimer as a predictive marker for mortality in patients with COVID-19 hospitalized during the first wave in Italy” by Shermarke Hassan et al., aims to demonstrate the value of D-dimer as a predictive marker for in hospital mortality in COVID-19 patients. The manuscript addresses a simple and important question using a multicenter study involving a large sample size. The manuscript is well written.

There are some points that need to be addressed:

1. The statistical analyses need to be clearer.

Three prediction models were tested to show the causal relationship between D-dimer levels and in hospital mortality: a model containing only D-dimer as a predictor, a model containing

age, sex and comorbidities, and a model containing D-dimer, age, sex and comorbidities.

Question: How was the calibration performed for each of the models using each of the quartiles? The authors show the calibration plots in figure 2, but is not clear for these calibration plots which quartile of D-dimer are referring to? Would be worth showing the calibration plot for each of their models developed using each of the quartiles.

Reply: Thank you for the insightful questions, we did not clearly mention how D-dimer was modeled in these analyses, which may have caused some confusion in understanding the method of analysis. When assessing the predictive value of D-dimer for in-hospital mortality (by looking at model discrimination and model calibration), we modeled D-dimer as a continuous variable, because this yields the best predictive value. (as categorizing a variable always leads to some reduction in predictive power) So figure 2 (the calibration plot) consists of prediction models that use D-dimer as a continuous variable. When assessing the association between D-dimer and in-hospital mortality we wanted to model the variable in a way that was more understandable from a clinicians’ point of view. A statement like “the relative risk of in-hospital mortality increases by X% for every 1 ng/mL increase in D-dimer levels” is a little hard to understand and doesn’t give you any sense of what this means in clinical practice. Therefore, we decided to model D-dimer using quartiles of D-dimer levels for this analysis. We have modified the manuscript so that this is clearer:

“All information was obtained from electronic patient records, using a standardized case report form. The exposure of interest, D-dimer levels (expressed as ng/mL) was used as either a continuous variable or a categorical variable depending on the analysis of interest. When showing descriptive statistics and estimating the association between D-dimer levels and in-hospital mortality, D-dimer was converted to a categorical variable with 4 levels that correspond to the 1st, 2nd, 3rd and 4th quartile of D-dimer levels. This was done to make the results easier to interpret for the reader. D-dimer was analyzed as a continuous variable when assessing the predictive value of D-dimer levels at admission for in-hospital mortality. This is because categorizing a variable always leads to some loss in predictive power.” (page 4)

We also rewrote the part of the manuscript where we explain model calibration:

“Model calibration refers to the degree to which the risk of mortality predicted by a model and the actual observed mortality rate in a group of patients are similar. For example, if the predicted mortality risk for a group of 100 patients is equal to 12% than the observed proportion of patients that died should also be around 12%. If the predicted mortality risk is much higher or lower than the observed mortality rate than the model is miscalibrated. To assess model calibration, we first fitted a model to the data, and then calculated the individual predicted mortality risk for each patient using this model. We then divided the patient population into ten groups (deciles) based on their predicted mortality risk. For each group, the predicted mortality risk for the whole group was compared with the observed mortality rate in that same group. This was done visually in a scatterplot that showed the predicted mortality risk on the X-axis and the observed mortality on the Y-axis for each decile. Furthermore, to examine calibration across the whole range, we also fitted a LOWESS (Locally Weighted Scatterplot Smoothing) line to the data.” (page 5/6)

Reviewer 2

The topic of the paper is timely and of clinical interest. Yet, I found the article to contain some incorrect statements and linguistic and grammatical errors. Thus, the article requires a complete revision. Specific comments:

Reply: Thank you for the important feedback, I have addressed your comments point-by-point below:

Question: Page 2, section “Results”: add percentage of number of patients with evaluable data

Reply: Done

Question: Page 3, section “Introduction”: numbers on people affected by COVID-19 and who died from the infection should be updated to April 2022

Reply: Done

Question: Page 3, section “Introduction”, sentence “it was important to understand the role of early predictive markers”: please specify. Predictive of mortality? Disease worsening? Disability? Long-term consequences?

Reply: Changed to “it was important to look for biomarkers measured at admission that could predict mortality and other in-hospital adverse outcomes, in order to better triage patients.”

Question: Page 3, section “Introduction”: please rephrase the sentence “thereby decreasing the burden on the healthcare system” and avoid brackets

Reply: Changed to “Patients predicted to have a low enough mortality risk wouldn’t need to be hospitalized, thereby decreasing the burden on the healthcare system.”

Question: Page 3, section “Introduction”, sentence “The secondary aim of this paper”: please remove the words “of this paper”, as they are a repetition

Reply: Done

Question: Page 3, section “Methods-Study design and population”: please specify the starting time point of follow-up period: follow-up was of 30 days starting from hospital admission? Viral infection confirmed by PCR result?

Reply: This was already mentioned in page 3 in the following way “The study population consisted of patients aged > 18 years who were hospitalized and who were positive for COVID-19 based on real-time PCR at 9 Italian hospitals, during the first COVID-19 wave in Lombardy, Italy (Feb-May 2020).

Question: Page 3, section “Methods-Study design and population”: please rephrase the sentence “In cases where it was not possible to obtain informed consent […] assuming the patient’s consent.” Grammar is not correct; please avoid the full stop between the words “death” and “Data collection”

Reply: Done

Question: Page 4, section “Methods-Data collection and definition of variables”: please remove the sentence “The primary endpoint […] mortality”, as endpoints have already been previously described

Reply: I have left this sentence in and modified it to “The outcome used for all analyses was in-hospital mortality” as it is the first time the outcome is mentioned in the methods section.

Question: Page 4, section “Methods-Data collection and definition of variables”: please define the Charlson comorbidity index, add reference

Reply: Added reference and changed text to “Lastly, the number of comorbidities (continuous variable) that each patient had at admission was calculated. Comorbidities to be assessed were selected based on their usage in the Charlson comorbidity index [8], a well-known risk score used to predict 10-year survival in patients with several comorbidities. To this list of comorbidities, we also added obesity (as defined by the clinician). The final list of comorbidities used was as follows: cardiovascular disease, chronic obstructive pulmonary disease, chronic kidney disease, diabetes mellitus, cancer, liver disease, dementia, connective tissue disease, acquired immunodeficiency syndrome (AIDS) and clinician-defined obesity.”

Question: Page 4, section “Methods-Data collection and definition of variables”, sentence “The total list of comorbidities was as follows;”: please change semicolons with colons

Reply: Done

Question: Page 4, section “Methods-Data collection and definition of variables”, words “HIV aids”: please specify HIV infection and AIDS

Reply: Changed to “acquired immunodeficiency syndrome (AIDS)”

Question: Page 4, section “Statistical analysis, general approach”: please define abbreviations before using them (e.g. SD, IQR…)

Reply: Done

Question: Page 4, section “Statistical analysis, general approach”, sentence “in that it (practically) precludes”: please rephrase avoiding the use of brackets

Reply: Removed brackets, changed “practically” to “almost completely”

Question: Page 4, section “Statistical analysis, general approach”, sentence “patients with missing values for the exposure”: please remove “the”

Reply: Done

Question: Page 5, section “Statistical analysis, the predictive value of D-dimer for in-hospital mortality”: please replace semicolons with colons

Reply: Done

Question: Page 5, section “Statistical analysis, the predictive value of D-dimer for in-hospital mortality”, paragraph “Model calibration was measured […] line to the data”: the whole paragraph is not clear and it is hard to follow. Could you please explain it better?

Reply: I rewrote the paragraph as follows: “Model calibration refers to the degree to which the risk of mortality predicted by a model and the actual observed mortality rate in a group of patients are similar. For example, if the predicted mortality risk for a group of 100 patients is equal to 12% than the observed proportion of patients that died should also be around 12%. If the predicted mortality risk is much higher or lower than the observed mortality rate than the model is miscalibrated. To assess model calibration, we first fitted a model to the data, and then calculated the individual predicted mortality risk for each patient using this model. We then divided the patient population into ten groups (deciles) based on their predicted mortality risk. For each group, the predicted mortality risk for the whole group was compared with the observed mortality rate in that same group. This was done visually in a scatterplot that showed the predicted mortality risk on the X-axis and the observed mortality on the Y-axis for each decile. Furthermore, to examine calibration across the whole range, we also fitted a LOWESS (Locally Weighted Scatterplot Smoothing) line to the data.”

Question: Page 5, section “Results – Baseline characteristics”, sentence “either anticoagulant […] and hospital admission”: please rephrase this sentence, as English grammar is not correct

Reply: I changed the sentence to “47 patients were excluded due to missing data for one or more confounding factors”

Question: Page 5, section “Results – Baseline characteristics”, sentence “between March 6th 2020 to September 6th 2020”: please replace TO with AND

Reply: Done

Question: Page 5, section “Results – Baseline characteristics”, sentence “Almost all […] COVID-19 wave”: please rephrase avoiding the use of brackets

Reply: Changed the sentence to “Almost all (98%) patients were enrolled before May 31st 2020, which roughly corresponds to the first three months of the initial COVID-19 epidemic in Italy”

Question: Page 6, section “Results – Baseline characteristics”, sentence “Who died doing so”: please rephrase

Reply: Changed to “75% of deaths occurred in the first 10 days”

Question: Page 6, section “Results – Baseline characteristics”: could you please specify how quartiles are defined and which are D-Dimer levels?

Reply: See the following text on page 4: “All information was obtained from electronic patient records, using a standardized case report form. The exposure of interest, D-dimer levels (expressed as ng/mL) was used as either a continuous variable or a categorical variable depending on the analysis of interest. When showing descriptive statistics and estimating the association between D-dimer levels and in-hospital mortality, D-dimer was converted to a categorical variable with 4 levels that correspond to the 1st, 2nd, 3rd and 4th quartile of D-dimer levels. This was done to make the results easier to interpret for the reader. D-dimer was analyzed as a continuous variable when assessing the predictive value of D-dimer levels at admission for in-hospital mortality. This is because categorizing a variable always leads to some loss in predictive power.”

Question: Page 7, section “Discussion”, sentence “The mechanisms underlying this COVID-19 induced coagulopathy”: please remove “the” and add a “-“ between COVID-19 and induced (COVID-19-induced coagulopathy”;

Reply: Done

Question: Page 7, section “Discussion”, sentence “The mechanisms underlying this COVID-19 induced coagulopathy may […] by monocytes”: add a reference explaining mechanisms of other bacteria-induced septic coagulopathies

Reply: Added reference

Question: Page 7, section “Discussion”, sentence “measured within 3 days of admission”: please substitute “of” with “from”

Reply: Done

Question: Page 8, section “Discussion”, sentence “9.1% of patients were on anticoagulant use […] during the study”: please rephrase avoiding the repetition of “anticoagulant use”

Reply: Changed to “). At baseline, 9.1% of patients were on anticoagulants and no information was given about thromboprophylaxis use during the study.”

Question: Page 8, section “Discussion”: please define abbreviations: VTE, DIC, TMA…

Reply: I have written them all out and removed the abbreviations as they are only mentioned 1-2 times.

Question: Page 8, section “Discussion”, sentence “For example in Lombardy […] were already being prescribed anticoagulant treatment”: please rephrase

Reply: Changed sentence to “For example, prophylactic anticoagulation for hospitalized COVID-19 patients became much more common over time.”

Question: As suggested also by Authors, an important limitation of this study is the absence of D-Dimer levels in more than 500 patients included in the study, which represent approximately half of the analyzed cohort. Moreover, D-dimer is a non-specific marker of inflammation, therefore it is likely to be high in COVID-19 infected patients, regardless of the presence of coagulopathy and/or infection-related thrombotic event.

Reply: You are correct, D-dimer is indeed also a marker of inflammation and therefore not specific to coagulopathies. For our purposes this was actually a good thing as we were interested in using D-dimer to predict overall mortality, not just thrombotic events. So the predictive value of D-dimer in this paper was based on its ability to act as a proxy measurement for 1) problems in coagulation as well as 2) general inflammation, which are both factors that lower the survival rate.

---

## [Editor Report · Decision Letter 1]

29 Jun 2022

The usefulness of D-dimer as a predictive marker for mortality in patients with COVID-19 hospitalized during the first wave in Italy.

PONE-D-22-03309R1

Dear Dr. Hassan,

We’re pleased to inform you that your manuscript has been judged scientifically suitable for publication and will be formally accepted for publication once it meets all outstanding technical requirements.

Kind regards,

Massimo Filippi

Academic Editor

PLOS ONE
---

## [Editor Report · Acceptance letter]

13 Jul 2022

PONE-D-22-03309R1 

The usefulness of D-dimer as a predictive marker for mortality in patients with COVID-19 hospitalized during the first wave in Italy. 

Dear Dr. Hassan:

I'm pleased to inform you that your manuscript has been deemed suitable for publication in PLOS ONE. Congratulations! Your manuscript is now with our production department. 

Kind regards, 

on behalf of

Prof. Massimo Filippi 

Academic Editor

PLOS ONE